# Lidar Line Selection with Spatially-Aware Shapley Value for Cost-Efficient Depth Completion

**Kamil Adamczewski**[1,2]   **Christos Sakaridis**[1]   **Vaishakh Patil**[1]   **Luc Van Gool**[1,3]

[1]Computer Vision Lab, ETH Zürich   [2]MPI-IS   [3]KU Leuven

kamil.m.adamczewski@gmail.com, {akamil, csakarid, patil, vangool}@vision.ee.ethz.ch

**Abstract:** Lidar is a vital sensor for estimating the depth of a scene. Typical spinning lidars emit pulses arranged in several horizontal lines and the monetary cost of the sensor increases with the number of these lines. In this work, we present the new problem of optimizing positioning of lidar lines to find the most effective configuration for the depth completion task. We propose a solution to reduce the number of lines while retaining up-to-the-mark quality of depth completion. Our method consists of two components, (1) line selection based on the marginal contribution of a line computed via the Shapley value and (2) incorporating line position spread to take into account its need to arrive at image-wide depth completion. Spatially-aware Shapley values (SaS) succeed in selecting line subsets that yield a depth accuracy comparable to the full lidar input while using just half of the lines.

**Keywords:** lidar, depth completion, feature selection, Shapley value

## 1   Introduction

Lidars have become indispensable for outdoor applications such as autonomous cars. They provide highly accurate range information at a fairly dense resolution compared to other active sensors such as radars. Moreover, their results are quite independent of the degree to which the surrounding scene is illuminated. The range information from a lidar provides a valuable signal for depth completion, i.e., for the estimation of a high-resolution depth map from an input camera image along with the lidar measurements. In such a setting, the accuracy of the completed depth map depends highly on the density of the lidar measurements. For the commonly used spinning type of lidar, this density is determined by the number of pulses emitted by the sensor at each azimuth, which corresponds to the number of horizontal scanning lines that the measurements form.

The cost of lidar sensors goes up with the number of scanning lines. Hence, increasing the measurement density tends to increase the overall cost. Consequently, performing depth completion based on fewer lidar lines is desirable. A naive approach to select a subset of lidar lines is keeping lines at regular, spatial intervals, which is the standard in the industry. But is this an optimal set-up, or does it make sense to try to build hardware with a custom line set-up? And if so, how can one achieve the latter? These are the main questions that this work attempts to answer.

We argue in this work that equally spaced line selection ignores the non-uniformity of the depth profile in real-world scenes. In particular, certain parts of the scene contain higher-frequency structures than others, which implies that using more lidar lines for measuring these parts can support more accurate depth map completion. Then, the line selection could greatly affect the depth completion performance because different lines contribute a different amount of importance. This work elaborates on the algorithmic approach how the custom lines can be selected and placed, thus providing the case that lidars with custom lines can be a viable option for further academic and industry research.

6th Conference on Robot Learning (CoRL 2022), Auckland, New Zealand.

In this paper, we propose an adaptive non-uniform selection of lidar lines for depth completion. To accomplish this, we utilize the so-called Shapley value [1] from the area of feature selection. The computed Shapley value for each lidar line indicates its marginal contribution to the overall depth completion output. This allows our method to directly evaluate the importance of each line for this task and keep the lines which are deemed most important. A limitation of these basic variants is that they treat lines as an unordered set, even though adjacent lines exhibit higher correlation than lines that are far apart. To account for this, we propose a spatially-aware Shapley value (SaS) scheme, in which the line selection further takes into account the spatial configuration of the lines by enforcing their selection to exhibit a minimum degree of regularity.

Furthermore, we introduce two basic variants of the approach, which select lines either at a global dataset level or at a local image level. In the former case, the scanning lines are fixed for the sensor. The entire line selection process is carried out once and then the identified distribution can be used to build a custom non-uniform sensor. In the latter case, the selected lines may differ between any two given images.

We experimentally validate the proposed lidar line selection approach on the KITTI dataset [2] for depth completion. The results demonstrate that SaS substantially outperforms both straightforward baselines and the basic Shapley value variants described above. SaS can maintain a depth error comparable to the error when using the full set of 64 lines, with as few as 32 lines. Our findings suggest that accurate depth sensing is attainable even with a reduced number of lines, which can make the usage of lidars in mass-production sensor suites for autonomous cars more feasible.

## 2    Related Work

**Depth completion** from sparse lidar measurements and a single RGB image was addressed in [3], which used a single deep regression network to predict depth. This work showed the benefit of using even few lidar samples for the increase of prediction accuracy. An encoder-decoder architecture is proposed in [4] for handling both sparse lidar and dense image data without the need for validity masks for the former. The requirement for semi-dense depth annotations is alleviated in the self-supervised approach of [5]. Surface normals are leveraged as intermediate or additional representations for densifying depth in [6, 7]. A fusion strategy is used in [8] to incorporate the information from the RGB image and correct potential errors in the lidar inputs. A Bayesian approach is followed in [9] to assign a posterior distribution to the depth of each pixel, modeling the sparse lidar points via likelihood terms. Non-local neighborhoods are proposed in [10] to iteratively propagate depth values across the image, while graph-based neighborhoods for propagation are utilized in [11, 12] via graph convolutions. Spatially-variant convolutional kernels inspired from guided image filtering are used in [13] to adaptively fuse features from the lidar and the camera branch. A cascaded scheme for depth prediction is presented in [14], using two complementary branches to compute the final depth map. A transformer-based network is introduced in [15], dynamically exploiting context across camera and lidar. An unsupervised framework for depth completion is proposed in [16], which learns to complete depth from sequences of measurements by backprojecting pixels to 3D space and minimizing the photometric reprojection error induced by the predicted depth. Crucially, all aforementioned methods either assume the existence of a full lidar scan, typically with 64 scanning lines, which comes from a very expensive sensor, or use a random subset of the lidar points. However, efficiently sampling random points from a spinning lidar is not feasible, as the mechanics of the sensor restrict potential subsampling to the level of entire scanning lines. Our method takes this constraint into account and automatically selects a subset of lidar lines by computing the importance of each line for the accuracy of the predicted depth.

**Feature selection and importance estimation** focuses on the effects that representations involved in learned models (such as neural networks) have on the final output of the models. This analysis aims at explaining the inner workings of the models. For a comprehensive overview of recent works on interpretable machine learning, we refer the reader to [17] and focus here on instance-wise feature explanation, which is related to our work. One line of work in this area consists of feature-additive

methods, such as LIME [18], SHAP [19], and attribution-based methods [20]. These methods estimate the importance of deep features along the channel dimension. LIME makes an assumption of local linearity of the examined model and permutes the features of new samples to weigh them according to their proximity to the model. SHAP is more closely related to our work, as it also uses the Shapley value [1] to explain the model by computing the deviation a given data sample exhibits from the global average of each feature. Both these methods output a feature importance weight, similarly to our work. Another line of work includes feature selection methods, such as L2X [21] and INVASE [22]. These methods aim at identifying a subset of the original features of the network that yields a similar output to that corresponding to the full set of features. This selection is hard, in the sense that the decision whether to keep or neglect each feature is not associated with a continuous importance score but rather with a binary variable. This stands in contrast with our Shapley value-based approach, which assigns a continuous score to each feature.

## 3   Problem Formulation

### 3.1   Definition of a Line

In this work, we refer interchangeably to a line, a channel, a lidar line or a lidar scan line as defined below. Lidar lines are separate lidar channels (or scan lines) from the lidar point cloud, as provided by the KITTI dataset [2]. The point clouds were generated with a Velodyne HDL-64E lidar sensor. Given a point cloud, we denote a point of it by $P_i = (x_i, y_i, z_i) \in \mathbb{R}^3$. We denote the position of the sensor by $P_n$. We adopt the same coordinate system convention as in [2]: x-axis (front), y-axis (left) and z-axis (up). If we consider a horizontal plane including the sensor, then the elevation angle $\theta_i$ with respect to this plane for point $P_i$ is given by:

$$\theta_i = \arcsin\left(\frac{(P_i - P_n)_z}{|P_i - P_n|}\right). \tag{1}$$

We group the resulting angles based on details provided in the Velodyne HDL-64E manual. The lidar has a vertical field of view of $26.9°$ and its vertical angular resolution is $0.4°$ on average. Although there are 64 channels, only 42 lidar channels overlap with the camera frustum.

### 3.2   Line Selection

In our line selection strategy, two modes are distinguished, static (global) and dynamic (local). In the static setting, the set of selected lines is fixed across the entire dataset. In the dynamic setting, a different set of lines can be selected for every frame. The global setting relates to a sensor with fixed position where the pulses are always emitted towards the same direction. On the other hand, the dynamic setting allows for a changing position / orientation of the transmitters. The limited set of lines is made to change depending on the input scene, i.e. on an image to image basis.

The problem of finding the optimal subset of $k$ lines out of $n$ possible positions is exponential in nature and would require the search among $\binom{n}{k}$ subsets. Due to this computational complexity, we reformulate the problem and, instead of computing the optimal subset directly, we look for the most advantageous individual lines. Thus, we aim to create a ranking of the individual lidar lines. Then for a given ranking and for arbitrary $k$, the top $k$ from the ranking form could be selected as the desired outcome.

A crucial question then is how we create the ranking of the lines. Our approach is to find the lidar line which, when added to an existing set of selected lines, would cause the biggest increase in performance. That is, for an arbitrary subset of already selected lines, we search for the line the marginal contribution of which is the largest. We address this point in Sec. 4 by employing the concept of the Shapley value.

**Table 1:** Comparison of terminologies in game theory (left) and our formulation of lidar lines rankings (right).

| Game theory | Ranking in CNNs | Notation |
|---|---|---|
| player | line | $n$ |
| characteristic function | RMSE | $\nu$ |
| coalition | subset of lidar lines | $\mathcal{K}$ |
| grand coalition | set of all the lidar lines | $\mathcal{N}$ |
| coalition cost | increase in RMSE after removing some of the lines | $\nu(\mathcal{K})$ |

## 4 Method

We divide this section in three parts. First, we present the overview of how the Shapley values (SVs) are computed for the lidar lines. Shapley values are real numbers that quantify the average contribution of a line in completing the respective depth map. Secondly, we present the approximation of the Shapley values based on linear regression, which allows to compute the SVs by incorporating sampling. Thirdly, due to the nature of the depth completion problem, we incorporate the concept of space or line spread.

### 4.1 Game Theoretical Lidar Line Ranking

This section tackles the problem of quantifying the role of a single lidar line in creating a depth map. The importance of a lidar line is described as the improvement in the quality of the depth map when we include a given lidar line. In other words, given a set of lines, the contribution of a line is the difference between the performance of the existing set of lines and the performance of the set of lines plus the line in question. To this end, we employ a concept from coalitional game theory, the Shapley value, which precisely quantifies the line's importance as its average marginal contribution.

#### 4.1.1 Coalitional Game Theory

Let a lidar line be called a player, the set of all players $\mathcal{N} := \{0, \ldots, N\}$ the *grand coalition* and a subset of players $\mathcal{K} \subseteq \mathcal{N}$ a coalition of players. Subsequently, we assess the utility of a given coalition, i.e., of a given subset of lines. To assess quantitatively the performance of a group of players, each coalition is assigned a real number, which is interpreted as a payoff or a cost that a coalition receives or incurs *collectively*. The value of a coalition is given by a *characteristic function* (a set function) $\nu$, which assigns a real number to a set of players. A characteristic function $\nu$ as defined before maps each coalition (subset) $\mathcal{K} \subseteq \mathcal{N}$ to a real number $\nu(\mathcal{K})$. Therefore, a coalition game is defined by a tuple $(\mathcal{N}, \nu)$.

In our case, a coalition is a subset of lines and the characteristic function evaluates the performance, which in our case is simply the root mean square error (RMSE) for the depth map produced based on the given subset of lines with respect to the ground-truth depth map. While in our case we define this characteristic function as the RMSE, it could be simply replaced by another metric such as mean absolute error (MAE), photometric error, etc.

Up until this point, we have defined the payoff given to a group of lines. There remains the question of how to assess the importance of a single line given the information about the payoffs for each subset of lines. To this end, we employ the concept of the Shapley value about the normative payoff of the total reward or cost.

### 4.2 Shapley Value

The concept introduced by Shapley [1] is a division payoff scheme which splits the total payoff into individual payoffs given to each separate player. These individual payoffs are then called the Shapley

values. The Shapley value of a player $i \in \mathcal{N}$ is given by

$$\varphi_i(\nu) := \sum_{\mathcal{K} \subseteq \mathcal{N} \setminus \{i\}} \frac{1}{N \binom{N-1}{|\mathcal{K}|}} (\nu(\mathcal{K} \cup \{i\}) - \nu(\mathcal{K})). \tag{2}$$

The value $\varphi_i(\nu)$ then quantifies the contribution of the $i$-th player to a target quantity defined by $\nu(\mathcal{N}) - \nu(\emptyset)$, that is the output of the characteristic function applied to the grand coalition minus the output when applied to the empty set. The sum over the Shapley values of all actors is equal to this target quantity, $\nu(\mathcal{N}) - \nu(\emptyset) = \sum_{i=0}^{N} \varphi_i(\nu)$. In our case, the grand coalition is the set of all the lines and the empty coalition corresponds to the case where no lidar measurements are used. Using the Shapley symmetrization ensures that the contributions are estimated in a "fair" way, i.e., according to a mathematically rigorous division scheme that has been proposed as the only measure that satisfies four normative criteria regarding fair payoff distribution. We describe these criteria in the Appendix.

### 4.3 Approximation via Weighted Least-Squares Regression

The Shapley value approximation describes the sets as binary vectors, where the vector dimensionality is equal to the total number of players. Each binary vector indicates whether a line is present in the respective subset or not. This allows to formulate Eq. 2 as a weighted least-squares regression problem. Nevertheless, since the exact Shapley value could only be obtained using exponentially many binary vectors, we resort to sampling. Given the subset $\mathcal{K}$, we create a binary vector $\mathbf{v}$ s.t. $|\mathbf{v}| = N$, $\nu(\mathbf{v}) = \nu(\mathcal{K})$ and

$$\mathbf{v}_i = \begin{cases} 1 & \text{if } i \in \mathcal{K}, \\ 0 & \text{otherwise.} \end{cases}$$

Alternatively, we can also sample the binary vectors directly by assigning $1/2$ probability to be either $0$ or $1$ to each vector entry or sample the vector based on the probability $1/\binom{N}{K}$ for a subset of length $K$. Consider then the Shapley values $\varphi_0(\nu), \dots, \varphi_N(\nu)$ to be the weights of the binary vector $\mathbf{v}$. As stated in [23], a formulation in this form allows to obtain the Shapley values as the solution of

$$\min_{\varphi_0(\nu), \dots, \varphi_N(\nu)} \sum_{\mathcal{K} \subseteq \mathcal{N}} \left[ \nu(\mathcal{K}) - \sum_{j \in \mathcal{K}} \varphi_j(\nu) \right]^2 k(\mathcal{N}, \mathcal{K}), \tag{3}$$

where $k(\mathcal{N}, \mathcal{K})$ are called the Shapley kernel weights which are defined as $k(\mathcal{N}, \mathcal{K}) := \frac{(|\mathcal{N}|-1)}{\binom{|\mathcal{N}|}{|\mathcal{K}|}|\mathcal{K}|(|\mathcal{N}|-|\mathcal{K}|)}$, where $k(\mathcal{N}, \mathcal{N})$ is set to a large number due to the division by 0. In practice, the minimization problem in Eq. 3 can then be solved by solving a weighted least-squares regression problem, the solution of which is

$$\phi = (\mathbf{V}^T \mathbf{K} \mathbf{V})^{-1} \mathbf{V} \nu,$$

where $\mathbf{V}$ is a matrix consisting of the above defined binary vectors, $\mathbf{K}$ the Shapley kernel weight matrix, and $\nu$ is a vector with the outcomes of the characteristic function applied to the corresponding subset in $\mathbf{V}$.

### 4.4 Space Constraint for Improved Depth Completion

The depth profile of a real-world scene is characterized by strong correlations at a local spatial level. Thus, these spatial correlations play an important role in predicting depth values in the setting of depth completion from sparse lidar measurements.

The role of the Shapley values is to create a ranking of lines without any additional constraints. As we will present in the experiments in Sec. 5, the lines identified as most important via the Shapley values may be in close spatial proximity, which leaves other regions of the scene without any depth information, even though such information would greatly facilitate depth completion in those regions. Therefore, the spatial structure of the set of lidar lines plays a key role in selecting lines, which motivates us to adapt our approach as follows in order to take it into account.

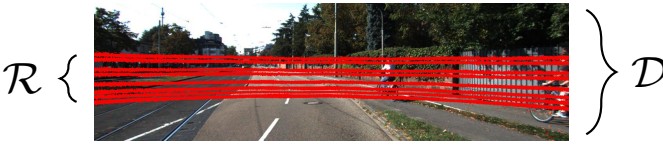

$\mathcal{R} \Big\{$ $\Big\} \mathcal{D}$

**Figure 1:** Visualization of regions described in Sec. 4.4. We denote $\mathcal{D}$ as the entire depth map and $\mathcal{R}$ as the minimal contiguous region that contains all the $N$ lidar lines and $S$ spread lines.

Let the spatial extent of the depth map $\mathcal{D}$ be divided into $D \in \mathbb{N}$ sub-regions which describe the lidar lines such that $|\mathcal{D}| = D$. Let $N \in \mathbb{N}$ be a fixed budget of lines. Then the lines which are neglected are $K = D - N$. Importantly, we note that the $N$ lines can be "spread out" or huddled together. For example, when the $N$ lines are all consecutive, they take up the space of one joined region of size $N$ (with the remaining space also being joined in one or two regions). However, this is exactly the case which we wish to avoid, as the depth measurements are not spread out over the entire scene.

Consider the minimal contiguous region of the image which contains all selected lidar lines and denote it by $\mathcal{R}$. This area $\mathcal{R}$ may contain regions corresponding to lidar lines that are not selected, which we describe as spread, $\mathcal{S}$. That is, $\mathcal{S} \subset \mathcal{R}$. The entire depth map $\mathcal{D} = \mathcal{R} \cup \mathcal{R}'$. When all the lines are huddled together, $|\mathcal{R}| = N$, $|\mathcal{R}'| = K$ and $|\mathcal{S}| = 0$. When there is one line that is not selected between the selected lines, then $|\mathcal{R}| = N + 1$ with $|\mathcal{S}| = 1$, and $|\mathcal{R}'| = K - 1$.

In the experiments of Sec. 5, we show empirically how increasing the spread $S$ improves the quality of the depth map, even when the number of lines remains constant.

## 5 Experiments

For the experimental validation of our method, we use the KITTI benchmark. KITTI consists of driving scenes recorded from the car. The data consist of RGB images, sparse range measurements produced by a lidar, and dense depth maps which are annotated as ground truth-depth images. To measure the performance of the depth completion task, we use the RMSE between the ground-truth depth and the predicted depth. Given a sparse set of depth points, we perform depth completion using the network proposed in [5].

We first train the network from scratch for 10 epochs. Subsequently, we select a subset of the lines according to a given method. We distinguish several ways to select lines. "Shapley global" is a static scheme, where the input in every frame consists of the same set of lines selected by the Shapley value. Shapley global is computed based on samples from the entire dataset. On the other hand, "Shapley local" performs the selection for every image separately (as given in Sec. 3). Here, to compute the SVs, multiple samples (350 in this case) are drawn for every image.

The second component of our method is the spatially-aware sampling strategy. The basic selection baseline selects equally spaced lines, starting from the top line (which is most significant according to our tests), e.g., line 64, 48, 32 and 16 for the case of $l = 4$, where $l$ is the number of selected lines. Subsequently, we combine the two components in two distinct variants. The first one, "SaS constant-$k$" fixes the least amount of space between the lines, that is any two selected lines need to be separated by at least $k$ intermediate lines. In our experiments, we set $k = 1$. Then SaS constant-1 describes a set of lines selected with the Shapley ranking with the spatial constraint that two lines cannot be adjacent to each other. The second variant, which we name "SaS flexible-$s$" proves to be the most effective. Here, spread budget of size $s$ is flexibly assigned between the signal lines. The details are given in the following.

Fig. 2 summarizes the performance of the variants described above for a range of line budgets. The number of the lines at our disposal influences the trade-off between the Shapley value of the selected lines and the degree of spatial coverage associated with them and determines in large part the method and the potential performance. As Fig. 2 shows, when the number of selected lines is larger than $50\%$ of the total number of lines, the preserved lidar signal is dense and spatial constraints are not very important, since the sheer number of lines suffices to obtain an accurate result. This is also

| Method / # of lines | 32 | 16 | 8 | 4 |
|---|---|---|---|---|
| Shapley (local) | 946 | 3215 | 5656 | 6620 |
| Shapley (global) | 1078 | 3872 | 5471 | 6848 |
| Spaced | 1149 | 1951 | 4289 | 7671 |
| Random | 1799 | 2416 | 5043 | 10490 |
| SaS constant-1 (local) | 1361 | 1419 | 3183 | 5838 |
| SaS flexible (global) | **888** | **1178** | **2147** | **3436** |

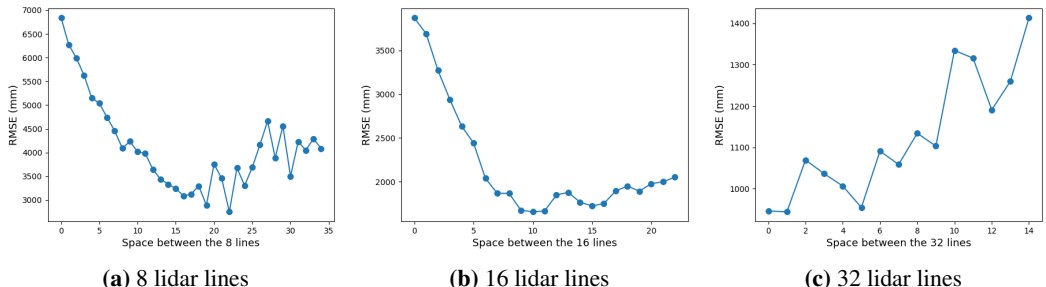

**Figure 2:** Performance comparison of lidar line selection methods for depth completion on KITTI. The table on the left shows the RMSE (in mm) for the compared methods with varying number of selected lines. The plot on the right delves deeper into the comparison of two of the methods, Shapley (local) and SaS constant-1 (local), comparing them for all possible numbers of lines. The RMSE with 64 lines is 853 mm.

**(a)** 8 lidar lines      **(b)** 16 lidar lines      **(c)** 32 lidar lines

**Figure 3:** The effect of including space between the selected lines on the quality of depth completion in terms of RMSE between the ground-truth and the predicted depth. For a fixed number of lines $N \in [8, 16, 32]$, the total spatial spread of them is varied (horizontal axis). Spread is described via $S$, the number of lines which are located between the two extreme selected lines and are not selected. For varying $S \in [0, 64 - N]$, $S$ random lines are selected and removed from the range of lines $[64 - N - S, 64]$ (the average over 15 samples). Note that the topmost line is numbered as 64.

confirmed by Fig. 3, which studies SaS flexible and the role of space in depth. In the case of 32 lines, on average, the RMSE rises as the spread increases. In the table in Fig. 2 we include the best sample which includes the amount of spread equal to 4 (the details of the exact spread configurations are given in the Appendix). In general, though, when the number of available lines is $> 32$, the best option is to select the lines directly produced by the Shapley value. Let us note that with 32 lines both SaS and the local Shapley value reach similar performance to the original RMSE with 64 lines which is 853 mm.

As the number of selected lines decreases, the relative importance of incorporating spatial awareness in our method increases. This importance is evident from the fact that simply selecting equally spaced lines yields better results than the Shapley value. The Shapley value itself selects lines which do not have adequate spatial coverage of the image and therefore do not produce a completed depth map of good quality. On the contrary, for small numbers of lines, incorporating a spatial spread constraint into the Shapley value scheme produces the best results. For 16 and 8 lines, SaS substantially improves the performance of both of the plain Shapley value variants and the equally spaced baseline, however one should note that when the number of available lines drops toward a one-digit number, it is hard to expect to match the result of the 64 lines. On the other hand, as the results show, even just a quarter of the lines produces a result which is acceptable. The visual results are presented in Fig. 4.

## 5.1 The Effect of the Spatial Constraint

As argued in Sec. 4.4, the space or spread $\mathcal{S}$ between the input features plays a role in depth completion. In this experiment we verify the role of space for the varying amount of spatial spread

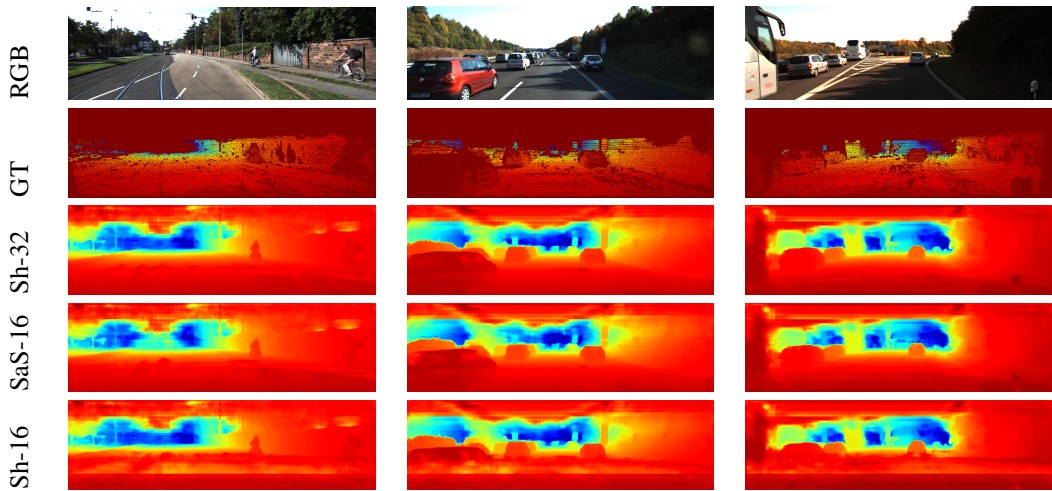

**Figure 4:** Examples of depth maps generated with the proposed methods with a subset of lidar lines. Shapley local (Sh-32 and Sh-16) uses 32 and 16 lines, respectively without the space constraint. SaS-16 uses both the Shapley value and the space constraint. Notice the role of space when using 16 lines, that is smoother prediction over the entire depth map when utilising the space constraint. Also notice more accurate prediction of the nearby car/bus in the bottom-left corner. When using 32 lines, denser input yields more accurate predictions both for closer and further regions without the need to incorporate spatial constraints. With fewer lines, Shapley focuses more on the far-away regions.

which is a simplified version of the "SaS flexible" method. Given a budget of $N$ lines, we verify the impact of the spatial extent of the region $\mathcal{R}$ as described in Sec. 4.4 by varying the size of $\mathcal{R}$. In the experiment, we fix $N$ to $4, 8, 16$ or $32$, and we vary $|\mathcal{R}| \in [N, 42]$ (42 is the number of visible lines). Since there are $\binom{R}{N}$ possible configurations, we randomly select indices for the lidar lines and verify the accuracy of the depth map inferred with the respective configuration of selected lines. The random sampling is repeated 15 times. The results are shown in Fig. 3.

Fig. 3 presents consistent results. As the number of available lidar lines increases, the role of spatial spread decreases. In the case of 32 lines, on average, including space actually deteriorates the result. Also in the case of 8 and 16 lines, empirically, the improvement from spreading the lines is noticeable up to a certain tipping point. Afterwards, as the number of empty lines increases, the quality of depth map decreases, as the signal becomes too sparse compared to the space between the lines.

SaS flexible differs from SaS constant in that we sample the top lines as given by the Shapley value ranking with higher probability assigned to lines which are ranked higher. As a result, for different samples, the amount of spread varies. Thanks to the empirical observations from this section, given a fixed number of lines, we allow for a "spread budget" and test only the samples which do not exceed the budget. This allows to sieve through the line configurations, making less samples necessary to obtain the same level of performance, as presented in Fig. 2.

## 6   Conclusion

Motivated by lowering the costs of access to lidar for widespread applications, we introduce the problem of lidar line selection for depth completion. Thanks to the algorithmic blend of the game-theoretical concept of the Shapley value and the spatial constraint necessitated by spatial scene correlations, our depth completion method predicts depth maps of comparable quality with only a fraction of the lines. These global line configurations can be used to build a custom non-uniform sensor. In the future, we aim to further improve the results for instantaneous local selection to reduce the computational overhead required at every frame. Morever, as we present the benefit of optimizing the selected lines, we expect the design of future lidar sensors will start being influenced accordingly.

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
