# OpenReview forum: "Lidar Line Selection with Spatially-Aware Shapley Value for Cost-Efficient Depth Completion"
_robot-learning.org/CoRL/2022/Conference — CoRL 2022 Poster_

### Official Review · Reviewer_3dHW · 2022-07-21

**Originality:** Good
**Technical Quality:** Good
**Clarity Of Presentation:** Good
**Impact:** 4

**Recommendation:**

Weak Accept: I recommend accepting the paper, but will not argue for my recommendation if the majority of other reviewers have a different opinion.

**Summary:**

This paper proposed a solution to select the most effective LiDAR lines for the depth completion task. Major contributions are as follows:

(1) The author proposed to leverage the Shapley value for LiDAR lines to evaluate the marginal contribution to the overall depth completion output.
(2) A spatially-aware Shapley value (SaS) scheme is proposed to take into account the spatial configuration of the line selection.
(3) The proposed method has two basic variants of the approach, the global version which select lines at a dataset level and the local version which selects lines for each image independently.

**Issues:**

**Major Issues:**
* The limitation section is missing.
* Analysis of computation cost is needed.
**Minor Issues:**
* The author could detailly state how the proposed method could contribute to the real-world application. If the line selection requires pre-training or sampling on dense LiDAR data, the value of the proposed method will need further justification.
* There is more SOTA depth completion methods that could be discussed in the Related Works section like:

[1] X. Cheng, P. Wang, C. Guan, and R. Yang. CSPN++: learning context and resource aware convolutional
spatial propagation networks for depth completion. AAAI Press.

[2] Choi, Jaehoon, et al. "Selfdeco: Self-supervised monocular depth completion in challenging indoor environments." 2021 IEEE International Conference on Robotics and Automation (ICRA). IEEE, 2021.

[3] Feng, Ziyue, et al. "Advancing self-supervised monocular depth learning with sparse liDAR." Conference on Robot Learning. PMLR, 2022.

[4] Yoon, Sungho, and Ayoung Kim. "Balanced depth completion between dense depth inference and sparse range measurements via kiss-gp." 2020 IEEE/RSJ International Conference on Intelligent Robots and Systems (IROS). IEEE, 2020.




**Quality Of The Limitations Section:**

Limitations section not present

**Reviewer Expertise:**

4: The reviewer is confident but not absolutely certain that the evaluation is correct

**Robotics Focus:**

Highly relevant to robotics but no hardware experiments

**Strengths And Weaknesses:**

**Strengths:**
* This paper proposed a novel method to approximate the marginal contribution of a LiDAR line to the depth completion task, which could provide valuable information for the sparse LiDAR design.
**Weaknesses:**
* In my understanding, the proposed method needs pre-training on the dense (64-line) LiDAR data before the line selection, which hinders its application on Sparse-LiDAR-only systems.
* The computation cost analysis is missing.
* It would be better if the line selection could be efficiently predicted only with the RGB image input, to guide the LiDAR design or selection.

**Summary Of Recommendation:**

The LiDAR line selection could greatly affect the depth completion performance because different lines contribute a different amount of importance. The problem author tried to solve is significant. Although the method and presentation are not perfect, it is still inspiring.

Post-rebuttal: the proposed method seems not practical for current rotation-based LiDARs, but it is still inspiring and may be possible to guide the design of future solid-state LiDARs which could generate variational scanning patterns. I would keep my rating of "weakly accept"

---

> ### Author Response · Authors · 2022-08-24
> **Response to Reviewer 3dHW**
>
> We appreciate the positive review and provide here further explanations.
>
> ### Applicability to different types of lidar measurements
>
> The proposed set-up indeed requires pre-training on the dense lidar information, which, however, makes the approach more general, as we can use the model for any set of lines, including sparse-lidar-only systems. We would only need to change the input layer for a smaller size and remove the appropriate weights from the first layer.
>
> ### Analysis of computational cost/complexity
>
> The computational cost of the method is the following. Let $s$ be the number of samples required to compute the Shapley value, and $f$ the cost of the forward pass. Then, the sampling cost is $sf$. Subsequently, the computation of the Shapley values requires solving linear regression. The complexity of this operation is $O(s^2p+p^3)$, where $s$ is the number of observations/samples and $p$ is the number of features. In our case, $p=64$. Additional cost comes from sorting the values, which is $O(p\log p)$ and is negligible for $p=64$. This process is performed only once in the global case, and for each frame/scene separately in the local case. The main computational burden is associated with collecting samples. For the global case, we are mostly unrestricted, as sampling should take place prior to deployment, but it should be restricted for the local case when the computation is done for the observed frame.
>
> ### Additional related works
>
> We will include further discussion in the Related Work section on the additional depth completion methods which were indicated by the Reviewer.

---

### Official Review · Reviewer_6id7 · 2022-07-31

**Originality:** Good
**Technical Quality:** Fair
**Clarity Of Presentation:** Good
**Impact:** 2

**Recommendation:**

Weak Accept: I recommend accepting the paper, but will not argue for my recommendation if the majority of other reviewers have a different opinion.

**Summary:**

This paper proposes a method for reducing the cost of accessing LiDAR for depth completion task. To achieve the saving, LiDAR lines are ranked with Shapley value. Lines with high Shapley values are considered important and kept while others are discarded.

**Issues:**

The objective and the overhead mentioned in the weakness section.

**Quality Of The Limitations Section:**

Limitations section not present

**Reviewer Expertise:**

4: The reviewer is confident but not absolutely certain that the evaluation is correct

**Robotics Focus:**

Highly relevant to robotics but no hardware experiments

**Strengths And Weaknesses:**

Strength:
- Method description is clear and easy to follow.
- Modeling the importance of the LiDAR line with Shapley value looks natural and also promising.
- Experiments for accuracy evaluation show the power of this method

Weakness:
- Abstract is hard to read
- The objective of this method is ambiguous. What cost is saved with your method?
  - Energy consumption for LiDAR? In my understanding, to assess the importance of the LiDAR lines, this method requires the data from all LiDAR lines. So no LiDAR lines can be completely shut down.
  - Computation for depth completion? As far as I know, most depth completion methods project the point cloud collected from LiDAR to a 2D plane. So no matter how many LiDAR points there are, the inputs to the network are always 2D planes with the same number of pixels. The cost for inference doesn't change.
- To assess the Shapley value, multiple samplings, inferences, and accuracy evaluations are required. The overhead of this sampling process seems to overwhelm the benefit of line selection, especially in the dynamic problem setting.

**Summary Of Recommendation:**

The modeling part and the result of this paper look good. Though not supported by current LiDAR, the method proposed in this paper does show the probability of cost saving. Though the objective and overhead are ambiguous in the main text, they are clarified during the rebuttal period. Thus I change my score and recommend accepting this paper.

---

> ### Author Response · Authors · 2022-08-24
> **Response to Reviewer 6id7**
>
> ### Clarification of the main objective of the paper
>
> The cost-effectiveness may indeed have different facets and we will clarify our objective. In this work, we focus on reducing the input size rather than the network size. The main motivation behind decreasing the number of the lines is that, as mentioned above, devices with fewer lines are substantially cheaper than those with more lines. Thus, the term "cost-efficient" in the title of the paper refers to the actual price of the lidar.
>
> As far as the computational cost is concerned, except for the input layer, we keep the size of the network the same, so the computational benefits are negligible. There exist methods such as pruning, quantization, or distillation which could be applied for improving network efficiency, however, this is not the focus of this work.
>
> ### Clarification on potential computational overhead
>
> The global line selection variant of our method induces *no overhead at inference time*, as it simply requires one forward pass which produces the depth map from the input given by the selected lines, which is the same when no selection is applied. The only overhead is at training time, which does not impact efficiency at inference time. For the local line selection variant, there is indeed an overhead, since we need to consider every frame separately. One possible solution is parallelization, as samples can be computed independently. The local case is generally a challenging case and definitely requires further research.

---

> > ### Comment · Reviewer_6id7 · 2022-08-25
> > **Reply to Authors**
> >
> > Thank you for your clarification.
> > In my understanding, your objective is to reduce the cost for LiDAR sensors, namely the price for LiDAR. However, this objective cannot be achieved with your method, at least at the current stage. Here is my reasons:
> > - For the static setting, the set of lines selected for initialization (the overhead part) is potentially different from the set of lines selected for the following steps.
> > - For the dynamic setting, line selections from frame to frame are different.
> >
> > To support all possible line selections, LiDAR must equip the union of line sets, which means that the number of lines required to be equipped in LiDAR is actually larger than the lines selected by your method.
> > For example, at the initialization step the method selects lines 0,1 and at the following steps the algorithm selects lines 2,3. It appears the method requires only 2 lines but actually the LiDAR must have 0,1,2,3 four lines.
> >
> > I think the fundamental problem comes from the constraint that scan line positions are fixed for rotation-based LiDAR, like in the previous example, we cannot change the line position from 0,1 to 2,3. But for other types of LiDAR like Livox or some soild-state LiDAR, the scanning position can be possibly changed. With these sensors, your method could possibly show its advantage. However, the scan patterns for these LiDARs are very different from the one you use in your paper, so I'll not change my score.

---

> > > ### Author Response · Authors · 2022-08-26
> > > **Response to Reviewer 6id7**
> > >
> > > Thank you for taking the time to reply to our response. We would like to further clarify the lidar line selection process here and justify why in the global (static) setting we can indeed reduce the number of fixed lines for the final rotation-based lidar sensor and thus reduce its price. In our global line selection, Shapley values for all lines (64 in the case of KITTI) are computed only once, at training time. Based on these Shapley values, a **fixed** non-uniform subset $K$ of the 64 lines is identified in the last step of our training pipeline. $K$ contains exactly the elevation values that are subsequently used to fixedly position the scanning lines of the new, cheaper rotation-based lidar sensor that is constructed based on the results of our method. The remaining lines that do not belong to $K$ are completely discarded after training and the corresponding elevation values are left empty in the new lidar sensor. Thus, the new lidar sensor includes a **reduced number of fixed lines**, which is indeed compatible with the technology of rotation-based lidars which does not allow variable line positions. As it was not possible for us to build such a sensor, we have experimentally verified its practical merit by "simulating" its readings, i.e., using only the fixed set of lidar lines that belong to $K$ for inference on KITTI and ignoring the rest of the lines at inference time. In this "simulation" of the new, cheaper sensor, we have shown that this sensor indeed retains almost the full performance of the original, expensive sensor for depth completion, achieving an RMSE of 0.888m with only 32 lines vs. the original RMSE of 0.853m with the full set of lines (see Fig. 2 on p. 7 of the paper). In the revised version of the paper which we have already uploaded, we have actually included the exact lists of IDs of the selected lines on KITTI (see Appendix II on p. 11 of the revised paper), which could be readily used for building the new, cheaper sensor (the corresponding elevation angles would be taken from the KITTI lidar specifications). We repeat these lists here for convenience:
> > >
> > > | Lines | RMSE | Configuration                                                                                   |
> > > |-------|------|-------------------------------------------------------------------------------------------------|
> > > | 4     | 3436 | 42-52-56-64                                                                                     |
> > > | 8     | 2147 | 39-46-51-55-57-62-63-64                                                                         |
> > > | 16    | 1178 | 35-36-39-44-47-48-49-52-54-55-56-58-59-61-62-64                                                 |
> > > | 32    | 888  | 27-28-29-30-33-34-35-36-37-39-41-43-44-45-46-47-48-50-51-52-53-54-55-56-57-58-59-60-61-62-63-64 |
> > >
> > > We indeed agree with the Reviewer that in the local (dynamic) setting, the approach described above for the global setting does not apply and further investigation to identify how local lidar line selection can be practically useful is required in future research. However, we note that our experiments have shown that global line selection, in particular the flexible spatially-aware Shapley value variant ("SaS flexible" in Fig. 2, p. 7), performs significantly better than all local line selection variants, which implies that the full potential of our lidar line selection approach is available for practical application to build cheaper lidar sensors, as we explained in the previous paragraph.
> > >
> > > We hope that this reply has fully backed the practical value of our method for constructing economical lidar sensors and that the Reviewer will consider it in their final rating of our paper.

---

### Official Review · Reviewer_xfBr · 2022-08-06

**Originality:** Good
**Technical Quality:** Fair
**Clarity Of Presentation:** Good
**Impact:** 2

**Recommendation:**

Weak Reject: I recommend rejecting the paper, but will not argue for my recommendation if the majority of other reviewers have a different opinion.

**Summary:**

The authors introduce the problem of selecting a subset of lines from a lidar scan that maximizes the accuracy of the depth map one obtains when completing the lidar line selection to compare against the original complete depth map. A solution to this line subset finding problem is presented which selects scan lines one by one based on their marginal contribution to the current line selection while encouraging spatial spread of the selected scan lines. The proposed method is evaluated on the KITTI dataset in conjunction with a deep-learning-based depth completion network. Various adaptations of the approach, such as various spatial sampling techniques and whether the line selection occurs once for all frames (global) or at every frame (local), are evaluated on the RMSE metric against the ground-truth lidar scans.

**Issues:**

# Major Issues

It seems that the appendix is missing for this submission. In line 164 a "supplement" is mentioned to describe the criteria of distribution fairness. In line 245 the authors again refer to an appendix.

Performance numbers should be provided for the given experiments. In particular for the local approach where the line selection happens at every frame, it is important to know whether the approach can run in real time. On that note, how many binary vectors were sampled to evaluate the Shapley values for the line selection candidates?

No limitations of the proposed method have been discussed in the paper.

# Minor Issues

It would be clearer to see how the method compares in Fig. 2 to only show the line diagram in Fig. 2 (right) without the table, and plot the evolution of the RMSE of all the methods.

Eq. 1 seems that it should not have the $\arg$ operator in it.

**Quality Of The Limitations Section:**

Limitations are not well addressed

**Reviewer Expertise:**

3: The reviewer is fairly confident that the evaluation is correct

**Robotics Focus:**

Relevant but unlikely to deploy to hardware in near future

**Strengths And Weaknesses:**

# Strengths

The approach is technically sound and explained clearly by introducing the Shapley value and the proposed spatial sampling techniques.

The evaluation of the method on the KITTI benchmark is detailed, with extensive ablations over the spatial sampling strategy and performance results, albeit only the RMSE is considered as accuracy metric.

# Weaknesses

The motivation for the introduced problem seems questionable to me. It is mentioned that "lowering the costs of access to lidar" (l. 280) motivated the proposed method. However, I do not see how it is economical to design new custom lidar hardware based on a non-uniform line pattern that has been found by this method when regular lidars measure equidistant scan lines. Do the authors have any evidence for the feasibility of lowering lidar cost through their approach?
Futhermore, when the line selection is allowed to change at every frame (Shapley local) or even between datasets -- what is the practical benefit/application of this? If the computing time can be reduced through less input data to perception pipelines, is the proposed method fast enough to warrant this additional lidar scan "downsampling" step at every frame? When we focus our attention to downsampling, how does Shapley-local compare to other approaches in this area, such as voxel-based downsampling?

Evaluating the method on a single dataset which has the same lidar placement accross all measurements potentially limits the generality of the proposed method. It would be interesting how the selection of lidar lines is affected by the tilt or height (for example) of the lidar, even on simulated data. Tying it back to the aforementioned practicality problem of designing custom lidar hardware -- is there any reason to assume that the same line selection will work globally under different conditions?

The paper does not discuss the choice of depth completion algorithm. Only in the experiment section is it mentioned that the approach from [5], a self-supervised deep learning method, is used as this crucial component which directly affects the performance of the overall pipeline. It should be explicitly said and better motivated in Sec. 2 that [5] is used as depth completion module. Furthermore, it has to be explained whether the network needs to be retrained for different line selections and what hyperparameters have been used. Do these hyperparameters influence which of the proposed line selection strategies yield the best results? If so, they must be included in the performance evaluations.

**Summary Of Recommendation:**

I cannot recommend acceptance of this work in its current form because it is difficult to assess the relevance to robotics of the introduced problem space and the proposed method. The authors should find a more convincing angle to motivate line selection (both global and local) and consider positioning the paper as an approach of downsampling of lidar scans, which is well motivated and has many well established baselines to compare against.

Futher issues to address to improve clarity and overall strength of the paper are given below.

---

> ### Author Response · Authors · 2022-08-24
> **Response to Reviewer xfBr**
>
> ### Motivation
>
> As mentioned in our general response, the main motivation in this paper is to propose a method which, unlike current lidars, uses a non-uniform line arrangement, and to verify that optimizing the line arrangement is beneficial in the first place. This fact is certainly not obvious, as uniform spacing of selected lines is a strong baseline based on our experimental comparisons, which drove us to incorporate spacing into our final SaS method.
>
> ### Choice of depth completion algorithm
>
> The upside is that the network does not need to be retrained for different line set-ups. It does not mean it cannot benefit from it. However, we wanted to keep the approach general and allow for possibility to change the line set-ups.
>
> ### Generality of the method across different lidar placements
>
> As an example of the influence of different lidar set-ups, we have carried out an analysis of different lidar height positionings, and we have added to the appendix the correspondences between lines and distances, where each line reaches for different heights of lidar placement. That said, as the height of different car models is highly concentrated around a certain value and lidars are typically mounted on the cars' roofs, the height of the lidar above the ground does not vary a lot across different settings.
>
> ### Minor issues
>
> Thank you for the very detailed read and for also pointing minor issues. All the missing details have been added to the new version.

---

### Author Response · Authors · 2022-08-24
**General comment to all Reviewers**

We would like to thank the reviewers for the thorough assessment of the paper with shrewd pointing to the potential weaknesses, which we will try to address below.

At the same time, we are glad that the reviewers consider the lidar line selection problem "significant" and deem the paper "inspiring", "technically sound", and "clear and easy to follow". Moreover, they assess the evaluation as "detailed", with "extensive ablations" and numerical results that "show the power of this method". We hope that this work with further improvements can make its way to the conference.

In this comment we address the main concerns which were shared by multiple reviewers, and provide individual comments to each reviewer discussing the specific points they raised. We also provide a revised version of our paper, which incorporates certain points that were indicated by the reviewers.

### Motivation and objective of the proposed approach

As far as the motivation and the objective of this work are concerned, lidars with uniformly spaced lines are the standard in the industry. But is this an optimal set-up, or does it make sense to try to build hardware with custom line set-up? And if so, how can one achieve the latter? These are the main questions that this work attempts to answer. As we have shown in the paper, different lines do not correspond to equally important features for depth completion. SaS focuses on the algorithmic approach which studies the problem whether attempting to build a non-uniform lidar sensor even makes sense.

Given the information available for the current lidars, there is a substantial price difference between sensors with different numbers of scanning lines. As an example, for the Velodyne lidars, the VLP-16 (Puck) costs about USD 4,000, and VLP-32 USD 20,000. Similarly, Robosense 16 and 32 sensors cost around USD 3,000 and USD 14,000 respectively. The increase in price as the number of lines increases is substantial and it would be much more economical to use a lidar with fewer lines. That said, creating custom set-ups would inevitably incur some extra cost, yet we currently have no more data to assess it. What the cost of non-uniform line set-ups is and whether they can be cheaper than uniform set-ups is an open question for the lidar industry.

Separately from this motivation, the problem of line selection can also be considered interesting from a purely theoretical perspective, providing insights on which parts of the scene are the most essential for depth completion.

### Practical value of the proposed approach

As far as the practicality of our approach is concerned, the final instantiation of the proposed method concerns the global selection case, where the scanning lines are fixed for the sensor. In this global case, the entire line selection process is carried out once and then the identified distribution can be used to build a custom non-uniform sensor. This implies that at inference time, which is the crucial stage for the practical deployment of robotic systems, our method incurs zero overhead for depth completion, thus preserving efficiency.

That said, local line selection indeed calls for further research, as it involves two main limitations compared to global selection. On the one hand, there is currently no lidar sensor which allows beams to have variable elevation depending on the scene. On the other hand, SaS requires sampling for every frame/scene to compute the optimal set of lines, which results in computational overhead (in this work, we sample a set of lines 50 times for each frame).

### Concluding remarks

To sum up, as soon as the benefit of optimizing the selected lines becomes clear (which is evidenced experimentally in this paper), the design of future lidar sensors will start being influenced accordingly. One should thus not necessarily confine the use of our method to what is possible today.

Once again, we appreciate all the comments and we will be happy to discuss and address any further concerns.

### Sources for cited prices

- Aliexpress
- https://www.robotshop.com/
- https://picclick.com/Velodyne-Lidar-Ultra-Puck-VLP-32-293362209748.html

---

### Meta-Review · Area_Chair_XXy3 · 2022-08-14

**Recommendation:** Accept (Poster)
**Confidence:** 4

**Metareview:**

Main strengths:
1. The paper is well written and the technical ideas are clear.
2. Experiments show that the method is effective for LiDAR line selection, but several cases/baselines/metrics of interest not tested (see below).

Main weaknesses:
1. Practical value of the method/target problem is not sufficiently justified.
2. Related to the above, the experiments are lacking thorough evaluation of the method (e.g., overheads) in the context of practical LiDAR systems.

After the rebuttal and discussions, the reviewers have largely kept their original ratings. There is a sense that the work is interesting enough to be accepted so as to spur future research on the topic, but the true usefulness of the the idea in the context practical LiDAR systems need to be further investigated.